# A preclinical model of chronic pancreatitis driven by trypsinogen autoactivation

Andrea Geisz[1] & Miklós Sahin-Tóth [1]

Inflammatory diseases of the pancreas have no specific therapy. Discovery of the genetic basis of chronic pancreatitis identified the digestive enzyme trypsin as a therapeutic target. Preclinical testing of trypsin inhibition has been hampered by the lack of animal models. Here we report the *T7D23A* knock-in mouse, which carries a heterozygous p.D23A mutation in mouse cationic trypsinogen (isoform T7). This trypsinogen mutant autoactivates to trypsin 50-fold faster than wild type. *T7D23A* mice develop spontaneous acute pancreatitis with edema, necrosis and serum amylase elevation at an early age followed by progressive atrophic chronic pancreatitis with acinar cell loss, fibrosis, dilated ducts and adipose replacement. Markedly elevated trypsin activity is apparent at first signs of pancreatitis and persists into later stages of the disease. This remarkable model provides in vivo proof of concept that trypsinogen autoactivation can drive onset and progression of chronic pancreatitis and therapy should be directed against intra-pancreatic trypsin.

---

[1] Center for Exocrine Disorders, Department of Molecular and Cell Biology, Boston University Henry M. Goldman School of Dental Medicine, Boston, MA 02118, USA. Correspondence and requests for materials should be addressed to M.S.-T. (email: miklos@bu.edu)

The inflammatory diseases of the pancreas comprise acute pancreatitis, recurrent acute pancreatitis, and chronic pancreatitis, which form a disease continuum and have no specific therapy[1]. Development of acute pancreatitis and subsequent progression to chronic pancreatitis is often promoted by mutations in risk genes that encode digestive proteases or their inhibitor. Pathogenic variants in *PRSS1* (cationic trypsinogen), *CTRC* (chymotrypsin C) and *SPINK1* (serine protease inhibitor Kazal type 1) increase conversion of trypsinogen to injurious trypsin either by stimulating autoactivation or by interfering with the protective mechanisms of trypsin inhibition by SPINK1 and trypsinogen degradation by CTRC[2]. Inappropriately high levels of trypsin activity in the pancreas cause acinar cell injury and consequent inflammation. The strongest disease-causing mutations in *PRSS1* are typically associated with hereditary pancreatitis. The clinically most frequent mutations (e.g., p.R122H, p.N29I, p.R122C, p.A16V) increase trypsinogen autoactivation by blocking CTRC-dependent trypsinogen degradation or by increasing CTRC-mediated processing of the trypsinogen activation peptide[2,3]. Importantly, a subset of mutations (e.g., p.D19A, p.D20A, p.D22G, p.K23R) that affects the trypsinogen activation peptide can robustly stimulate autoactivation in a CTRC-independent manner[4]. Despite the overwhelming genetic and in vitro biochemical evidence that support a direct pathogenic role for intra-pancreatic trypsinogen autoactivation caused by genetic mutations, confirmation from appropriate animal models has been lacking. More importantly, the absence of animal models that would develop spontaneous chronic pancreatitis driven by trypsinogen autoactivation hampered preclinical testing of therapeutics targeting intra-pancreatic trypsin. In this regard, a previous attempt to create transgenic mice with wild-type and mutated forms of human *PRSS1* (p.N29I and p.R122H) yielded a model in which disease penetrance was low, late-onset and independent of mutation status[5]. A similar transgenic model with *PRSS1* p.R122H showed no spontaneous phenotype[6]. Transgenic mice carrying p.R122H-mutated mouse T8 trypsinogen were described to exhibit features of acute and chronic pancreatitis, however, this promising strain was eventually lost to time[7]. Finally, a tamoxifen-inducible conditional Cre-driven transgenic strain carrying a furin-activated artificial trypsinogen construct was reported to develop acute acinar cell damage followed by fatty replacement[8]. Mechanistic interpretation of these results, however, has been confounded by potential Cre toxicity and the ambiguous properties of the transgene utilized. Taken together, none of the published mouse models has been suitable as a preclinical test tool for therapeutic trypsin inhibition. In the present study, we filled this knowledge gap by generating a novel knock-in mouse strain carrying a heterozygous p.D23A mutation in the activation peptide of the endogenous mouse cationic trypsinogen (isoform T7). Increased trypsinogen autoactivation in these mice gives rise to spontaneous pancreatic pathology that recapitulates key phenotypic features of human acute and chronic pancreatitis.

## Results and Discussion

**Effect of p.D23A mutation on T7 mouse trypsinogen in vitro.** To model chronic pancreatitis associated with increased trypsinogen autoactivation in the mouse, we set out to mutate the endogenous cationic trypsinogen (isoform T7) in a manner that increases its autoactivation. The mouse genome contains 20 trypsinogen genes and we demonstrated previously that in the resting pancreas only four trypsinogens are expressed to high levels; isoforms T7, T8, T9, and T20[9]. T7 constitutes approximately 40–50% of total trypsinogen and it autoactivates more rapidly and to higher trypsin levels than the other mouse isoforms[9–11]. To create a mutant T7 that would robustly increase autoactivation, we introduced an Ala mutation in place of Asp23 (p.D23A) in the activation peptide (Fig. 1a). We chose to mutate this position because our previous studies on hereditary pancreatitis-associated mutations showed that the analogous mutation p.D22G strongly stimulated autoactivation of human cationic trypsinogen (PRSS1) and this effect was independent of CTRC[4]. Note that amino-acid numbering in mouse T7 is shifted by one relative to human PRSS1 due to an additional Asp residue in the activation peptide. We used an Ala replacement instead of the Gly found in human patients to achieve a stronger phenotypic change. To test the activation properties of the T7 p.D23A mutant in vitro, we purified recombinant wild-type and mutant T7 trypsinogen and measured autoactivation at pH 8.0 in 1 mM calcium. We observed a dramatic 50-fold increase in autoactivation with the p.D23A mutant when compared to wild type (Fig. 1b). Because trypsinogen activation can be initiated by cathepsin B and trypsinogen degradation can be catalyzed by cathepsin L, we tested the effect of these lysosomal cysteine proteases on the p.D23A mutant. In sharp contrast to the huge effect observed with autoactivation, there was no change in cathepsin-mediated cleavages of the p.D23A mutant relative to wild-type T7 (Fig. 1c). This control experiment clearly confirmed that mutation p.D23A altered autoactivation while other trypsinogen activation and degradation pathways remained unaffected. Thus, mutation p.D23A was an ideal tool to test the effect of increased trypsinogen autoactivation in vivo.

**Generation of the *T7D23A* knock-in mouse.** A targeting vector containing the mouse T7 gene with the p.D23A mutation in exon 2 and a neomycin resistance cassette flanked by loxP sites in intron 1 was used to introduce the mutation into the mouse genome by homologous recombination in C57BL/6 embryonic stem cells (Fig. 1d). To obtain *T7D23A* mutant mice, the neomycin cassette was removed by breeding with a Cre-deleter strain. The final *T7D23A* strain harbored the p.D23A mutation in exon 2 and a residual loxP site in the neighboring intron 1. To assess the expression of the mutant T7 allele at the mRNA level, we prepared cDNA from the pancreas of 3-week-old mice with no pancreatic pathology. DNA sequencing of these heterozygous mice demonstrated two peaks at the site of the mutation, which were comparable in height indicating similar mRNA expression levels for the wild-type and mutant T7 alleles (Fig. 1e). Unchanged T7 expression in *T7D23A* mice was also confirmed at the protein level with the help of a newly developed polyclonal antibody specific for mouse T7 trypsinogen (Fig. 1f). The novel *T7D23A* strain showed no gross phenotypic alterations and bred normally. *T7D23A* mice were slightly smaller but gained weight with similar kinetics as the C57BL/6N controls up to 6 months after which their weight gain plateaued (Fig. 1g). Since homozygous mice were runted and died around 2 months of age, all subsequent experiments were performed with heterozygous mice. Both male and female mice were studied.

**Spontaneous acute and chronic pancreatitis in *T7D23A* mice.** Macroscopic pancreas morphology in the *T7D23A* mice was normal at the age of 2–3-weeks but pancreata were more variable in size at 4–5-weeks and then became visibly smaller after 2 months of age. Examination of hematoxylin-eosin stained pancreas sections from *T7D23A* mice showed normal pancreas histology at 2 weeks and in the majority of 3-week-old animals (Fig. 2a). Starting at 3 weeks and more frequently ( ~ 40%) observed at 4–5-weeks, *T7D23A* mice developed typical acute pancreatitis with edema disrupting the tightly packed tissue

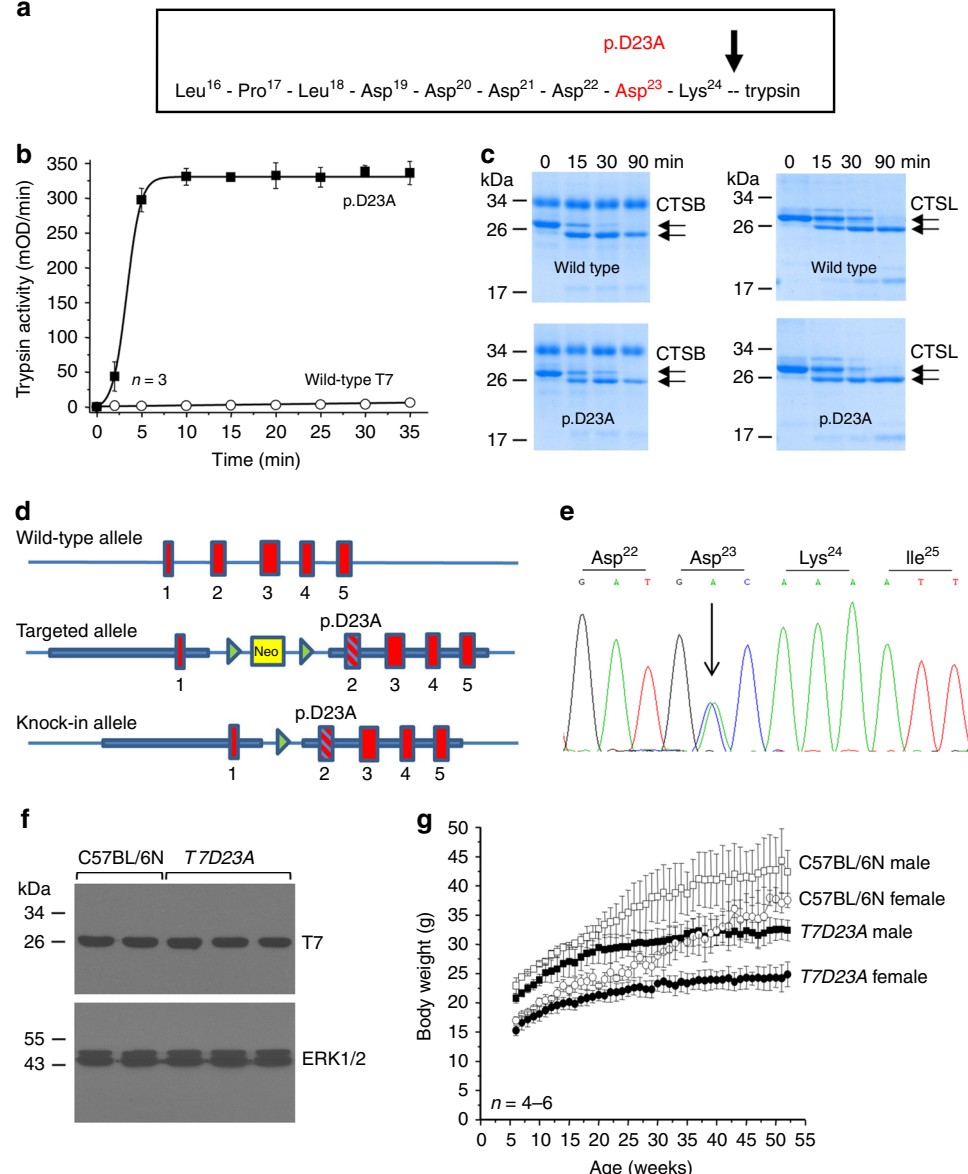

**Fig. 1** Effect of mutation p.D23A in T7 trypsinogen. **a** Location of mutation p.D23A (highlighted in red) in the activation peptide of mouse cationic trypsinogen, isoform T7. The arrow indicates the site of the proteolytic cleavage that activates trypsinogen to trypsin. **b** Autoactivation of purified wild-type (open circles) and p.D23A mutant (solid squares) T7 trypsinogen. Data points represent mean ± S.D. (n = 3). **c** Activation and degradation of purified wild-type and p.D23A mutant T7 trypsinogen by cathepsin B (CTSB) and cathepsin L (CTSL), respectively. CTSB cleaves after Lys24 and creates active trypsin. CTSL cleaves after Gly27 and generates an inactive product. The arrows point at the intact and cleaved trypsinogen bands. **d** Generation of the *T7D23A* knock-in mouse model. The targeting strategy used homologous recombination to create an allele carrying the p.D23A mutation in exon 2 and a neomycin resistance gene (Neo, in yellow) flanked by loxP sites (green arrowheads) in intron 1. The neomycin cassette was removed by breeding with a Cre-deleter strain. The numbered red boxes illustrate the five exons. The thick blue lines indicate the homology arms. **e** Expression of the p.D23A mutant allele. Pancreatic cDNA from 3-week-old heterozygous *T7D23A* mice was subjected to Sanger sequencing. The electropherogram indicates comparable peak heights at the site of the mutation for the wild-type (nucleotide A) and mutant (nucleotide C) alleles. **f** Expression of T7 trypsinogen protein in C57BL/6N and *T7D23A* 3-week-old mice. Western blot was performed with a T7-specific polyclonal antibody. ERK1/2 was measured as loading control. **g** Weight gain of male (square symbols) and female (circle symbols) C57BL/6N (open symbols) and *T7D23A* (solid symbols) mice. Data points represent mean ± S.D. (n = 4–6)

architecture, massive inflammatory cell infiltration, and centrally localized necrosis in the lobules (Fig. 2b). Immunohistochemistry (IHC) for myeloperoxidase (MPO) and F4/80 markers revealed that the inflammatory infiltrate consisted of neutrophil granulocytes and macrophages, with a slight preponderance of the latter (Fig. 3). In contrast, IHC for the T and B lymphocyte markers CD3 and CD45R/B220 was essentially negative (Fig. 3).

Disease progression was already evident in more than half of the 4–5-week-old mice studied, where pancreatic sections showed minimal intact parenchyma with widespread signs of regeneration including pseudotubular complexes, distorted, enlarged ducts, and diffuse interstitial fibrosis (early chronic pancreatitis, Fig. 2c). We confirmed fibrosis by Masson's trichrome staining (Fig. 4a) and by measuring elevated levels of hydroxyproline (Fig. 4b). Widespread activation of pancreatic stellate cells was also evident, as judged by IHC for the alpha-smooth muscle actin marker (Fig. 4c). Starting at 2 months and continuing through 6 and 12 months, adipose replacement

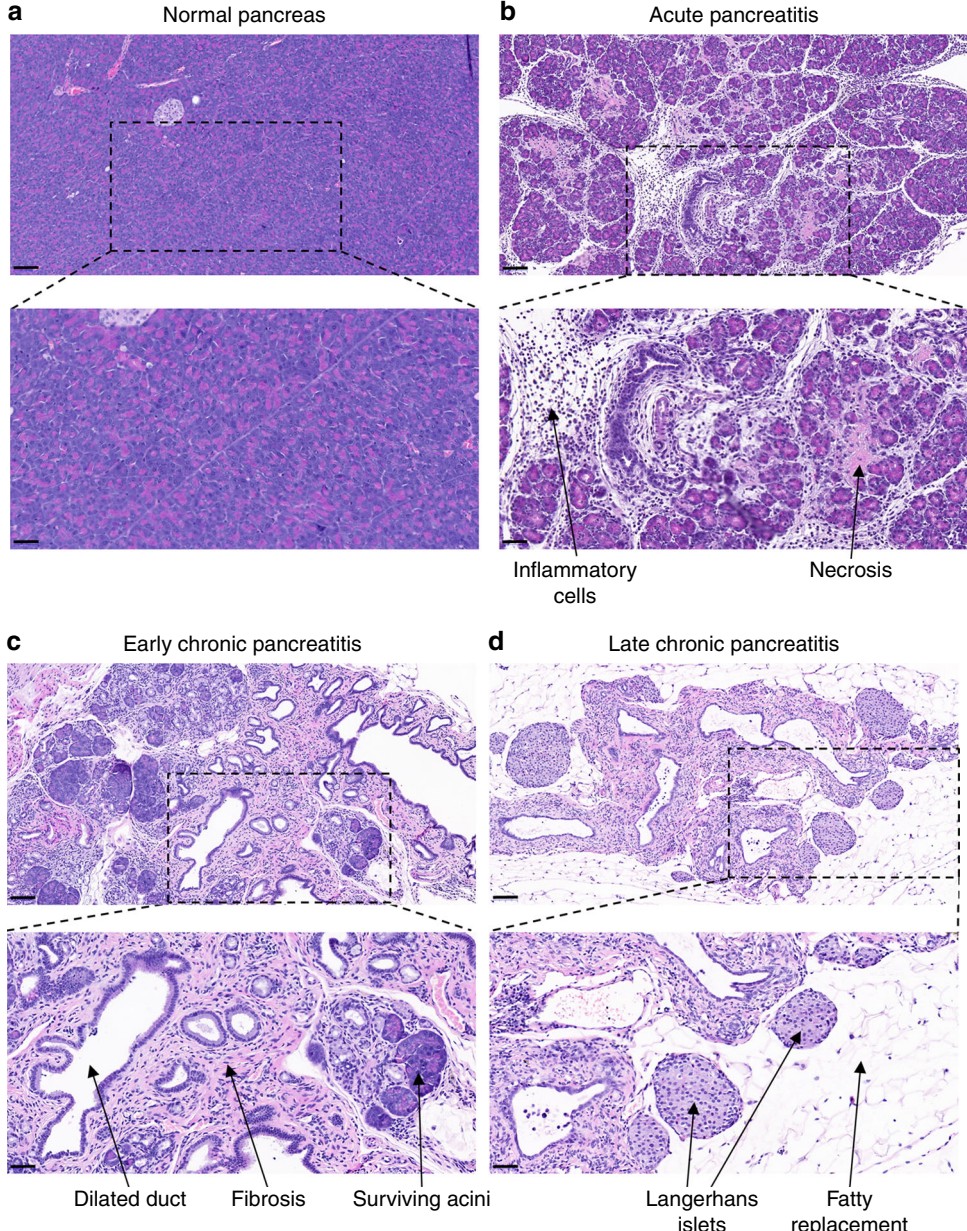

**Fig. 2** Pancreas histology in *T7D23A* mice. Hematoxylin-eosin stained pancreas sections are shown. **a** Normal pancreas from C57BL/6N control mice. **b** Acute pancreatitis was seen in 3–5-week-old *T7D23A* mice. **c** Early chronic pancreatitis was observed starting at 4 weeks. **d** Late chronic pancreatitis was evident after 2 months of age. See text for details. Scale bars correspond to 100 μm (upper panels) and 50 μm (lower panels)

began to dominate the histological picture while inflammatory cells and pseudotubular complexes became fewer in numbers. Dilated, distorted ducts surrounded by prominent fibrosis and sometimes filled with eosinophilic material, became more prevalent (late chronic pancreatitis, Figs. 2d and 4d). No calcifications were observed, based on negative von Kossa staining. Progression to the late phase was somewhat variable and a fraction of the mice showed mixed histology even at later ages. Notably, in the atrophied pancreas the islets of Langerhans remained abundant and appeared somewhat enlarged (Fig. 2d). Blood glucose measurements in *T7D23A* mice aged 12 months showed normal values ($114 \pm 35$ mg/dL, mean $\pm$ SD, $n = 5$) when compared to age-matched C57BL/6N controls ($113 \pm 42$ mg/dL, $n = 13$). We also reviewed hematoxylin-eosin stained sections of the lung, liver, spleen, and kidney from *T7D23A* mice (aged 6 weeks). In contrast to the pancreas, these major organs exhibited normal histology when compared to control C57BL/6N mice.

We quantitated the loss of acini and the appearance of adipose tissue on histological sections from a large number of *T7D23A* animals (Fig. 5a, b). Remarkably, disease penetrance was 100%. Acinar cell dropout started to occur at 4 weeks of age and the number of residual acini per visual field remained relatively stable below 30% between 2 and 12 months of age, with most mice having ∼ 10% or less intact parenchyma (Fig. 5a). Taking the marked pancreatic atrophy into account (see below), this would correspond to an even more severe acinar tissue loss when compared to a normal-sized pancreas. The time-line of adipose infiltration closely matched acinar cell loss but the extent was more variable with a wide distribution of fatty changes observed at ages 2, 6, and 12 months in the *T7D23A* mouse population studied (Fig. 5b).

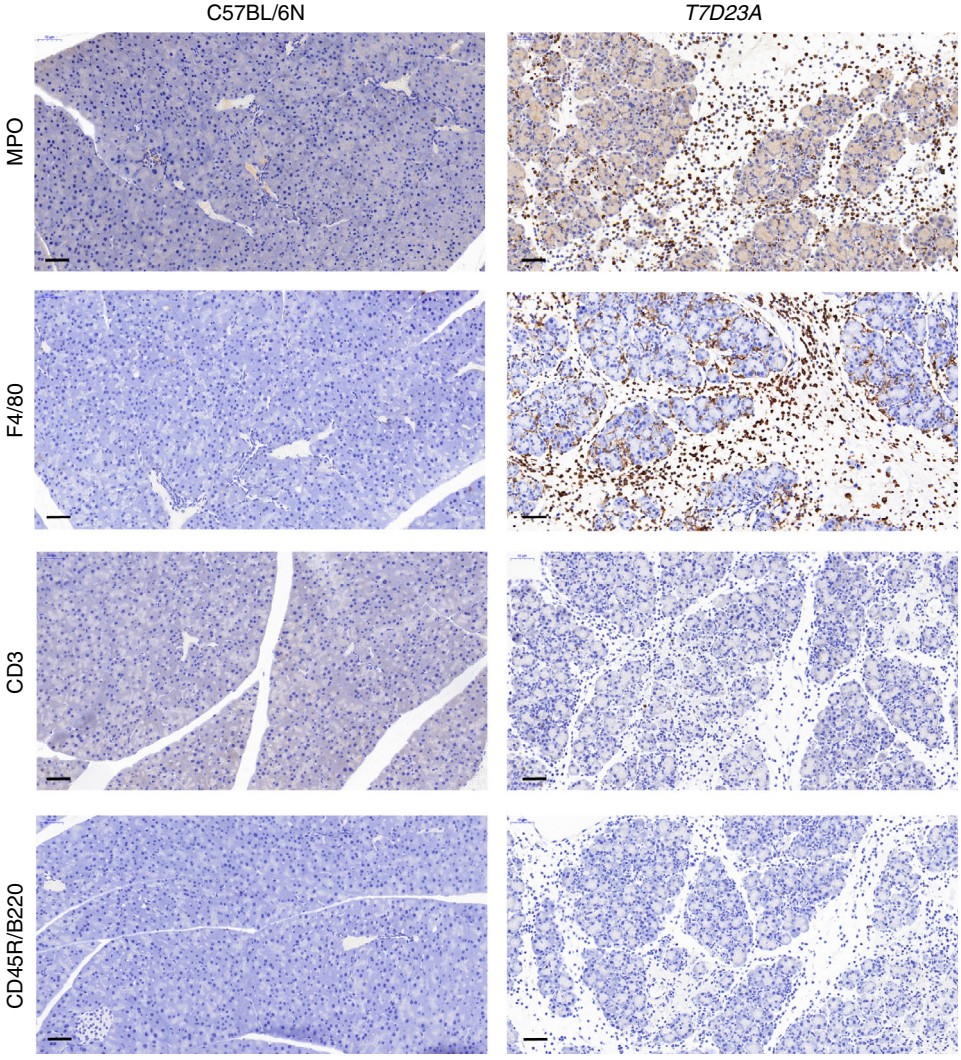

**Fig. 3** Immunohistochemistry for leukocyte markers in *T7D23A* mice. Pancreas sections from *T7D23A* mice with acute pancreatitis (aged 5 weeks) and C57BL/6N controls were stained for myeloperoxidase (MPO), F4/80, CD3, and CD45R/B220. Scale bar is 50 μm

**Pancreas weight in T7D23A mice**. When the pancreas weight of *T7D23A* mice was compared to that of control C57BL/6N mice, a marked increase was apparent in about 40% of the mice studied at 4–5-weeks of age (Fig. 5c). This weight gain was consistent with the edematous acute pancreatitis observed on histological slides. Reduced pancreas weight was evident in about 50% of the mice at 4–5-weeks, indicating early chronic pancreatitis. All *T7D23A* pancreata showed marked atrophy associated with late chronic pancreatitis at 2, 6, and 12 months of age, which corresponded to ~ 70% decrease in pancreas weight relative to those of control mice (Fig. 5c).

**Plasma amylase activity in T7D23A mice**. As a well-established serum marker of acute pancreatitis, we measured amylase activity from blood plasma of *T7D23A* mice at various ages (Fig. 5d). Remarkably, in about 40% of the mice tested, we observed high amylase values at 4–5-weeks; a sign of acute pancreatitis as already indicated by histology pictures and pancreas weight measurements. It is interesting to note that we were unable to capture acute pancreatitis in all mice studied; in all likelihood because of the weekly sampling times used and the variability in age of onset. However, it is also possible that not all mice develop acute pancreatitis; similarly to human patients who may present

with chronic disease in the absence of prior acute attacks. As expected, plasma amylase activity was lower in *T7D23A* mice than in control C57BL/6N mice at 6 and 12 months, consistent with loss of acinar cell function (Fig. 5d).

**Intra-pancreatic trypsin activity in T7D23A mice**. We measured trypsin activity from freshly prepared pancreatic homogenates of *T7D23A* mice at 3 weeks (no pathology), 4 weeks (acute or early chronic pancreatitis) and 2 months (late chronic pancreatitis) (Fig. 5e). When compared to age-matched C57BL/6 N mice, no elevation in trypsin activity was seen at 3 weeks, whereas high values were measured at 4 weeks. Significantly elevated trypsin activity was still observed at 2 months despite the drop in functional acinar cells; indicating that premature intra-pancreatic trypsin activation persist throughout the disease course. These observations, taken together with the fact that mutation p.D23A increases trypsinogen autoactivation, strongly support the conclusion that increased intra-pancreatic trypsinogen autoactivation can cause acute pancreatitis onset and subsequent progression to chronic pancreatitis.

**T7D23A,K24G mice develop no pancreas disease**. To exclude possible alternative disease mechanisms unrelated to trypsinogen

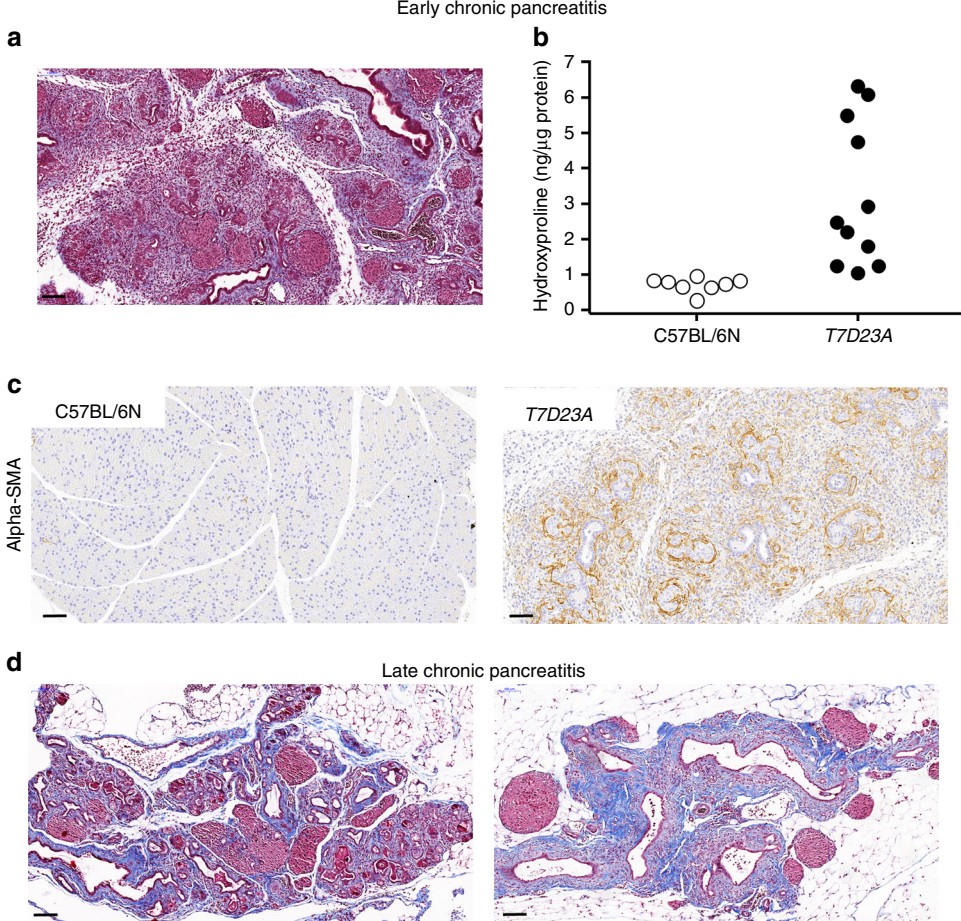

**Fig. 4** Fibrosis in the pancreas of *T7D23A* mice. **a** Masson's trichrome staining in early chronic pancreatitis (4 weeks). Connective tissue is stained blue. Scale bar is 100 μm. **b** Hydroxyproline content of the pancreas of *T7D23A* mice and C57BL/6N controls aged 4 weeks. **c** Stellate cell activation in the pancreas of *T7D23A* mice and C57BL/6N controls aged 4 weeks. Immunohistochemistry for alpha-smooth muscle actin (alpha-SMA). Scale bar is 50 μm. **d** Masson's trichrome staining in late chronic pancreatitis (left, 2 months, right, 12 months). Scale bar is 100 μm

autoactivation, we generated a mouse strain with a heterozygous *T7D23A* trypsinogen allele in which the trypsinogen activation site Lys24 was also mutated (p.D23A,K24G double mutant). Details of this strain will be reported elsewhere. Importantly, *T7D23A,K24G* mice sacrificed at the age of 5.5 months showed no spontaneous pancreas pathology. This observation offers compelling evidence that alternative pathological mechanisms such as trypsinogen misfolding or neoantigen formation are not responsible for the phenotype associated with the *T7D23A* allele and further supports the pathogenic role of increased trypsinogen autoactivation.

**Comparing features of *T7D23A* mice to those of human pancreatitis**. This new model recapitulates the clinical disease continuum of human pancreatitis in as much as it develops acute pancreatitis, which progresses to irreversible chronic pancreatitis. However, we did not observe recurrent episodes of acute attacks often seen in the human condition. This difference might be explained by the strong effect of the trypsinogen mutation used here, which drives earlier onset and rapid progression. Human chronic pancreatitis often leads to the end-stage pathology of parenchymal destruction with extensive fibrosis. Instead, in our case we typically observe adipose replacement of the atrophied acinar tissue. Fatty replacement is also seen in humans, mostly in conditions where relatively rapid acinar cell destruction occurs

(see[12] and references within). Importantly, adipose replacement of acinar cells was also reported in *PRSS1*-related hereditary pancreatitis, particularly earlier in the disease course[13].

A recently released mechanistic definition of human chronic pancreatitis provides a benchmark against which our model can be evaluated[14]. "Chronic pancreatitis is a pathologic fibro-inflammatory syndrome of the pancreas in individuals with genetic, environmental and/or other risk factors who develop persistent pathologic responses to parenchymal injury or stress." These criteria are completely fulfilled in our model. Disease onset and progression is clearly driven by a genetic change, which results in a persistent pathological response. In addition, "Common features of established and advanced chronic pancreatitis include pancreatic atrophy, fibrosis, pain syndromes, duct distortion and strictures, calcifications, pancreatic exocrine dysfunction, pancreatic endocrine dysfunction, and dysplasia." In the *T7D23A* model, we can observe many, but not all, of these salient features. Atrophy is prominent, ductal changes are widespread, and regenerative signs such as pseudotubular complexes are common. Although some fibrosis is seen early on, ultimately adipose replacement rather than severe fibrosis dominates the histopathology. We did not observe calcifications and there was no apparent indication of pain. Lack of ductal calcification is likely due to differences in ductal physiology in mice that limit bicarbonate levels while the lack of pain is probably related to the relative scarcity of fibrosis and the

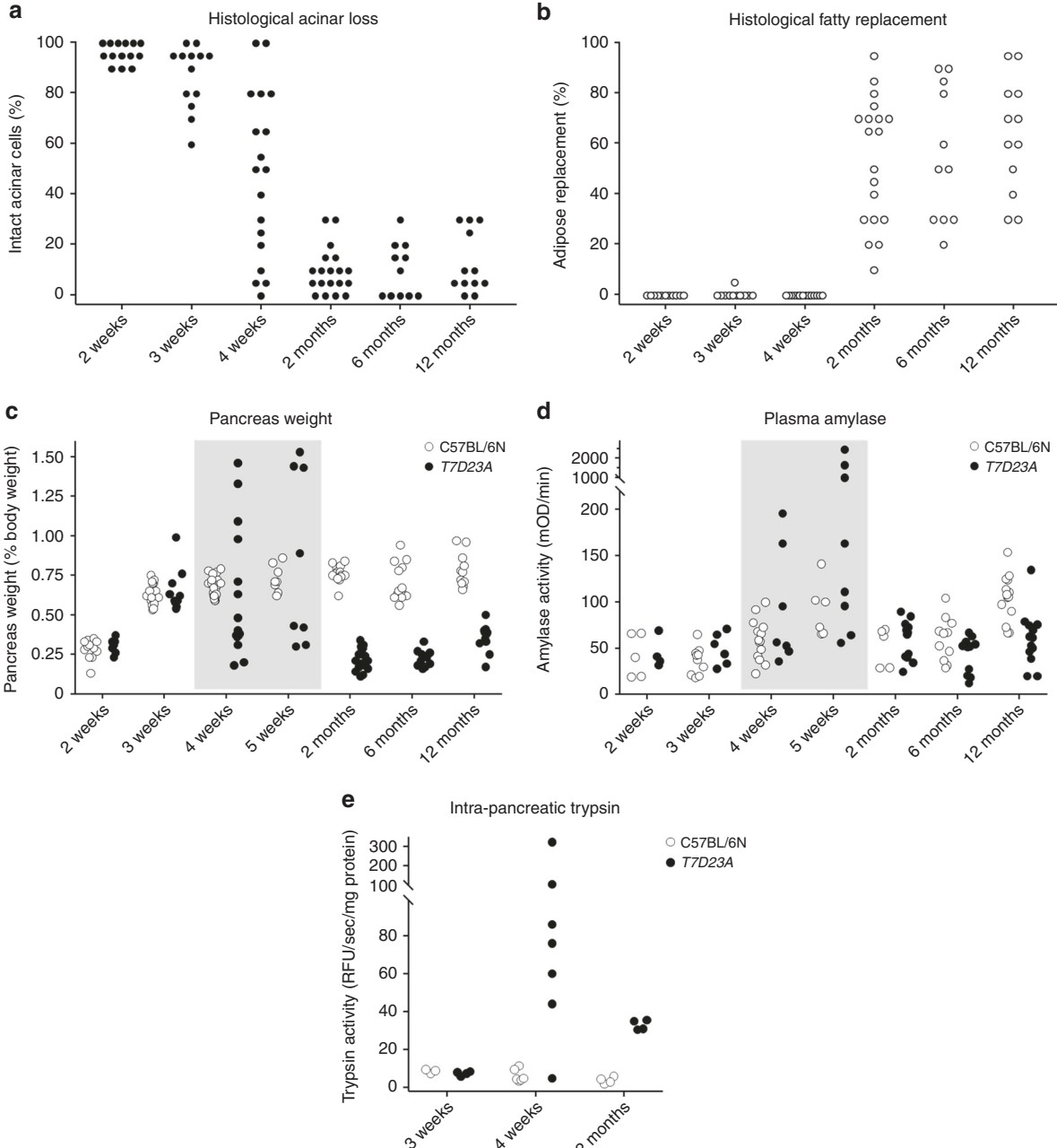

**Fig. 5** Pancreatic injury in *T7D23A* mice as a function of age. **a** Histological evaluation of acinar cell loss. Hematoxylin-eosin stained pancreas sections were visually scored for the presence of intact acinar cells. **b** Histological evaluation of adipose replacement. Hematoxylin-eosin stained pancreas sections were visually scored for the presence of fatty areas. **c** Pancreas weight of *T7D23A* (solid circles) and C57BL/6N (white circles) mice expressed as percent of body weight. The shaded area highlights the age when acute pancreatitis was detected most frequently. **d** Plasma amylase activity of *T7D23A* (solid circles) and C57BL/6N (white circles) mice. **e** Trypsin activity in pancreatic homogenates from *T7D23A* (solid circles) and C57BL/6N (white circles) mice

extensive fatty replacement in the end-stage pathology of the pancreas. We did not study exocrine dysfunction but we can infer this from the inability of older mice to gain weight, whereas we observed no signs of endocrine dysfunction.

In summary, we present here the first preclinical mouse model of chronic pancreatitis driven by inappropriately increased intra-pancreatic trypsinogen autoactivation. This model recapitulates salient features of human chronic pancreatitis associated with genetic mutations, including hereditary pancreatitis. Thus, the *T7D23A* mutant mice develop early-onset acute pancreatitis followed by progressive chronic pancreatitis. Remarkably, despite the dramatic pancreatic pathology, the mice breed well and

develop normally into adulthood, which is a prerequisite of their preclinical utility. At the conceptual level, the model provides direct in vivo evidence that trypsinogen autoactivation can drive onset and progression of chronic pancreatitis and treatment approaches should be directed against intra-pancreatic trypsin.

## Methods

**Accession numbers and nomenclature**. NC_000072.6, *Mus musculus* strain C57BL/6J chromosome 6, GRCm38.p4 C57BL/6J; NM_023333.4, *Mus musculus* RIKEN cDNA 2210010C04 gene (2210010C04Rik), mRNA; trypsinogen 7 (isoform T7). See Table 1 in ref. [9] for a complete list of mouse trypsinogen genes. Amino-acid residues are numbered starting with the initiator methionine of the primary translation product. Note that because of an extra Asp residue in the activation

peptide of T7, amino-acid numbering in this isoform is shifted by one relative to human trypsinogens.

**Animal studies protocol approval**. We have complied with all relevant ethical regulations. Animal experiments were performed with the approval and oversight of the Institutional Animal Care and Use Committee (IACUC) of Boston University, including protocol review and post-approval monitoring. The animal care program at Boston University is managed in full compliance with the US Animal Welfare Act, the United States Department of Agriculture Animal Welfare Regulations, the US Public Health Service Policy on Humane Care and Use of Laboratory Animals and the National Research Council's Guide for the Care and Use of Laboratory Animals. Boston University has an approved Animal Welfare Assurance statement (A3316-01) on file with the US Public Health Service, National Institutes of Health, Office of Laboratory Animal Welfare, and has been accredited by the Association for Assessment and Accreditation of Laboratory Animal Care International (AAALAC).

**Generation of the *T7D23A* mouse strain**. Mice were on the C57BL/6N genetic background. Mutation c.68A>C (p.D23A) was knocked-in to the mouse T7 gene (2210010C04Rik) using homologous recombination in C57BL/6 embryonic stem (ES) cells (Cyagen, Santa Clara, CA). The gene encoding T7 trypsinogen is located on chromosome 6; it spans ~ 3.8 kb and comprises 5 exons. The targeting vector contained the T7 gene with the p.D23A mutation in exon 2 and a 1924 nt sequence including a neomycin resistance gene flanked by loxP sites in intron 1, which served as a positive selection marker (Supplementary Fig. 1). Correctly targeted ES cell clones were identified by long-range PCR and confirmed by Southern blot. Mutant ES cells were injected into mouse embryos (blastocysts), which were implanted into pseudopregnant females. The resulting chimeras were bred with wild-type C57BL/6N mice to achieve germline transmission of the mutant allele. To remove the neomycin resistance gene from the mutant allele, mice were bred with a Cre-deleter strain that expresses the Cre recombinase in the early mouse embryo (B6.FVB-Tg(EIIa-cre)C5379Lmgd/J; Jackson Laboratories). The final *T7D23A* knock-in allele carried the p.D23A mutation in exon 2 and a 127 nt residual sequence in intron 1 containing a single loxP site (Supplementary Fig. 2). *T7D23A* mice were maintained in the heterozygous state by breeding with C57BL/6N mice obtained from Charles River Laboratories (Wilmington, MA). Littermate C57BL/6N mice were used as experimental controls. Both male and female animals were studied.

**Generation of the *T7D23A,K24G* mouse strain**. This control strain was created using the same methodology as described above for the *T7D23A* strain. Details on this strain will be reported elsewhere.

**Genotyping**. To genotype *T7D23A* mice, we used primers that amplified exon 2 with the flanking intronic sequences. The amplicon size from the wild-type allele was 630 bp, whereas the mutant allele yielded a 757 bp product due to the presence of the residual sequence in intron 1. The primer sequences were as follows. Forward: 5′- CTT GAA ACT AAC AGT GGA CCC T -3′; Reverse: 5′- AAC TGT GCA CAT TTC CTA ATT G -3′.

**Histology**. Pancreas tissue was fixed in 10% neutral buffered formalin; paraffin-embedded (FFPE); sectioned and stained with hematoxylin-eosin or Masson's trichrome staining at the Boston University Experimental Pathology Laboratory Service Core. Von Kossa staining was performed at the Specialized Histopathology Core, Brigham and Women's Hospital, Boston, MA.

**Immunohistochemistry (IHC)**. Staining for the leukocyte markers MPO, F4/80, CD3, and CD45R/B220 was carried out at the Specialized Histopathology Core, Brigham and Women's Hospital, Boston, MA. Formalin-fixed paraffin-embedded tissue samples were sectioned at 4 μm thickness. IHC staining was executed on a Leica Bond III Automated IHC Stainer (Leica Biosystems, Buffalo Grove, IL). Antigen retrieval was achieved by heating for 30 min in citrate buffer (pH 6.0) for F4/80, CD3 and CD45R/B220 and in EDTA (pH 9.0) for MPO. Primary antibody binding to tissue sections was visualized using Bond Polymer Refine Detection kit (catalog #DS9800, Leica Biosystems) except for CD45R/B220 where a goat anti-rat mouse-adsorbed IgG (Vector Laboratories, Burlingame, CA, catalog #PI-9401) was used at 1:50 dilution for 30 min. The 3,3′-diaminobenzidine tetrahydrochloride hydrate (DAB) substrate was employed to generate a brown precipitate at the sites of peroxidase activity. Cell nuclei were counterstained blue with hematoxylin. The following primary antibodies were applied at room temperature for 30 min. MPO, Dako (Agilent, Santa Clara, CA), catalog #A0398, rabbit polyclonal antibody, dilution 1:1000. F4/80, Cell Signaling Technology, catalog #70076, rabbit monoclonal antibody clone D2S9R, dilution 1:500. CD3, Cell Signaling Technology (Danvers, MA), catalog #99940, rabbit monoclonal antibody clone D4V8L, dilution 1:75. CD45R/B220, BD Biosciences (San Jose, CA), catalog #550286, rat monoclonal antibody clone RA3-6B2, dilution 1:200.

Staining for alpha-smooth muscle actin was performed by Reveal Biosciences (San Diego, CA). Heat-induced antigen retrieval was carried out for 20 min in

citrate buffer (pH 6.0). The primary antibody was from Abbiotec (San Diego, CA), mouse monoclonal antibody clone 1A4, catalog #251813. The BioCare Mouse on Mouse HRP-Polymer (BioCare Medical, Pacheco, CA, catalog #MM620H) was used for detection with DAB substrate and hematoxylin counterstain.

**Hydroxyproline determination**. Hydroxyproline content in the pancreas was determined by measuring the reaction of oxidized hydroxyproline with 4-(dimethylamino)benzaldehyde (catalog #AK008, Sigma-Aldrich, St. Louis MO), according to the manufacturer's instructions[15]. Values were normalized to total protein and expressed in ng hydroxyproline per μg protein units.

**RNA isolation and reverse transcription**. Total RNA was extracted from mouse pancreas ( ~ 30 mg tissue) using the RNeasy Plus Mini Kit (Qiagen, Valencia CA). RNA (2 μg) was reverse-transcribed using the High Capacity cDNA Reverse Transcription Kit (4368814, ThermoFisher Scientific).

**Western blotting**. A rabbit polyclonal antibody was raised against the peptide sequence LKTAATLNSRVST corresponding to amino-acids 114-126 of mouse T7 pre-trypsinogen (Li Scientific, Denver CO). The specificity of the antibody was validated using pancreas homogenates from a T7-deficient mouse[10,11]. A rabbit monoclonal antibody (catalog #4695) against p44/42 MAPK (ERK1/2) (137F5) was purchased from Cell Signaling Technology. For western blotting, 30–40 mg mouse pancreas tissue was homogenized in 300–400 μL ice-cold phosphate-buffered saline supplemented with Halt protease and phosphatase inhibitor cocktail (Thermo Scientific) and 20 μg homogenate was electrophoresed on 15% sodium dodecyl sulfate polyacrylamide gel electrophoresis (SDS-PAGE) mini gels and transferred onto an Immobilon-P membrane (Millipore, Bedford, MA). After blocking with 5% non-fat milk in PBS supplemented with 0.1% Tween 20, the membrane was incubated with the primary and secondary antibodies for 1 h each at room temperature. The bands were detected using SuperSignal West Pico Chemiluminescent Substrate (ThermoFisher Scientific). Antibody dilutions were as follows: anti-T7 1:10,000, anti-ERK1/2 1:1000, horse-radish peroxidase-conjugated anti-rabbit 1:10,000. Uncropped versions of western blots are provided in Supplementary Fig. 3.

**Intra-pancreatic trypsin activity**. The pancreas ( ~ 40–50 mg) was homogenized in 1 mL MOPS homogenization buffer (250 mM sucrose, 5 mM MOPS (pH 6.5), 1 mM MgSO₄), using a motorized homogenizer. The homogenate was briefly centrifuged (1000 × *g*, 1 min) and an aliquot (20 μL) of the cleared homogenate was mixed with 30 μL assay buffer (50 mM Tris-HCl (pH 8.1), 150 mM NaCl, 1 mM CaCl₂, 0.1 mg/mL bovine serum albumin) and 150 μL of 200 μM Boc-Gln-Ala-Arg-AMC fluorescent substrate (Bachem USA, Torrance CA) dissolved in assay buffer. The increase in fluorescence was followed for 3 min in a fluorescent plate reader at 380 nm excitation and 460 nm emission wavelengths. The rate of substrate cleavage was expressed as relative fluorescent units (RFU) per second and it was normalized to the total protein in the assay mix (RFU/s/mg protein).

**Plasma amylase**. Levels of amylase in the blood were measured using the 2-chloro-*p*-nitrophenyl-α-D-maltotrioside substrate (catalog #A7564-60, Pointe Scientific, Canton MI) in a kinetic activity assay. Plasma (4 μL) was diluted with 6 μL normal saline and mixed with 190 μL substrate to start the reaction. The increase in absorbance due to the release of 2-chloro-nitrophenol was monitored in a microplate reader at 405 nm for 2 min. The rate of substrate cleavage was expressed in mOD/min units.

**Expression plasmids and mutagenesis**. We constructed a new bacterial expression plasmid for mouse T7 trypsinogen in which the N terminus was fused to the C terminus of a self-splicing mini-intein (pTrapT7 intein-mouse T7 construct). During bacterial expression, the fusion undergoes spontaneous self-splicing giving rise to trypsinogen with an authentic, homogeneous N terminus[16]. The fusion construct was generated by gene synthesis (GenScript, Piscataway, NJ) and cloned in the pTrapT7 plasmid using *Nco*I and *Sal*I restriction sites[16]. Mutations p.D23A and p.S201A were introduced by overlap extension PCR mutagenesis. Double mutant p.D23A,S201A was constructed by cut-and-paste from the single mutant constructs.

**Expression and purification of mouse T7 trypsinogen**. Wild-type and mutant intein-trypsinogen fusions were expressed in the aminopeptidase P deficient LG-3 *Escherichia coli* strain[16]. Purification of trypsinogen was performed by ecotin affinity-chromatography[16]. To stabilize trypsinogen against autoactivation during purification, the 50 mM HCl elution solution also contained 100 mM NaCl[4]. Concentration of trypsinogen preparations was estimated from the ultraviolet absorbance at 280 nm using the extinction coefficient 39,140 M⁻¹ cm⁻¹.

**Trypsinogen autoactivation**. Mouse cationic trypsinogen (isoform T7) at 2 μM concentration was incubated with 10 nM initial trypsin in 0.1 M Tris-HCl (pH 8.0), 100 mM NaCl, 1 mM CaCl₂, and 0.05% Tween 20 (final concentrations) at 37 °C.

At the indicated times, 2 μL aliquots were withdrawn and trypsin activity was measured with 200μM N-CBZ-Gly-Pro-Arg-p-nitroanilide substrate.

**Trypsinogen cleavage by cathepsin B and cathepsin L**. Activation and degradation of T7 trypsinogen by cathepsins was studied on catalytically inactive (p.S201A mutant) T7 constructs to exclude the confounding effect of autoactivation. Trypsinogen (1 μM) was incubated at 37°C in 0.1 M Na-acetate (pH 4.0), 100 mM NaCl, and 1mM K-EDTA with 37 μg/mL recombinant human cathepsin B[17] or 7.1 μg/mL human liver cathepsin L (219402-25UG, Calbiochem) (final concentrations). Cathepsins were activated with 1 mM dithiothreitol before use. Reactions (100 μL) were terminated by precipitation with 10% trichloroacetic acid and analyzed by 15% SDS-PAGE and Coomasie Blue staining.

## Data availability

All data generated or analyzed during this study are included in this published article and its supplementary information files.

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

## Acknowledgements

This work was supported by Department of Defense grant W81XWH-14-1-0331 and National Institutes of Health (NIH) grants R01 DK058088 and R01 DK117809. We are grateful to Balázs Csaba Németh, Eszter Hegyi, Anna Orekhova, and Zsanett Jancsó for technical assistance and helpful discussions.

## Author contributions

M.S.-T. conceived and directed the study. A.G. and M.S.-T. designed the experiments. A.G. performed the experiments. A.G. and M.S.-T. analyzed the data. M.S.-T. wrote the manuscript with significant input from A.G.

## Additional information

**Competing interests:** M.S.-T. is a consultant for Takeda Pharmaceuticals. A.G. declares no competing interests.

