## [Peer Review File · Nature Communications]

Reviewers' comments:

Reviewer #1 (Remarks to the Author):

The submission by Geisz and Sahin-Toth characterizes a new murine model of chronic pancreatitis that is designed to parallel mechanistically and pathologically the human disease known as Hereditary Pancreatitis. This genetic etiology of this disease is mutation of the human cationic gene; to recapitulate this disease in mouse, the authors have made a knock-in a murine trypsinogen isoform that they predict will reflect the defect that causes human disease. Previous attempts to generate such a mouse model have had limited success for a variety of reasons. This model is proposed to cause disease by enhancing the autoactivation of the trypsinogen. The authors show that the mutation induces prominent pancreatic disease; further characterization is needed to determine to what extent it parallels human disease. The input of a pathologist familiar with the pancreatitis, especially the histologic features of Hereditary Pancreatitis as well as murine models of pancreatic disease would be useful.

1. The authors state that the earliest changes in pathology are those of acute pancreatitis. However, the pathology shown appears not to show acute inflammation, a central feature of acute pancreatitis. Indeed, the inflammation appears to be primarily mononuclear. As part of the initial characterization of the model, the authors should label the tissue for inflammatory cell subtypes (neutrophils, lymphocytes, and macrophages) and quantify the infiltrates at various disease stages. If lymphocytes are abundant, particularly at the very early stages of disease, they should be subtyped (at least as T or B categories). Since the authors propose this model might be used for clinical trials, it is critical to establish the similarity (and differences from human disease).
2. In the context of #1, the dramatic mononuclear response, especially at the early stage of disease, seems very unusual. I wonder if the authors have generated a form of lymphocytic pancreatitis and if the mutated trypsinogen could be a neo-antigen. In this context, the authors should compare serum from WT, early, and late time period mutants (several animals at each age), for antibodies to general pancreatic determinants (homogenates) as well as to the mutant protein.
3. In that context, an elevation in serum levels of amylase does not necessarily mean that pancreatitis is present. For example, changes in duct permeability and/or basolateral exocytosis could cause similar findings. If there are no neutrophils to accompany the elevations in amylase, this reviewer would caution about the pathologic term acute pancreatitis, even if the amylase is elevated.
4. One of the hallmarks of Hereditary pancreatitis is intraductal calcifications-some images suggest these may be present-it would be of value for the authors to review the pancreatic pathology with an pathologist, with one goal of being to state whether there are intraductal calcifications. Moreover, some centers have high resolution ultrasonography and/or computed tomography for mice. It would be very useful to show pancreatic calcifications by imaging in addition to pathology.
5. The mutant protein has a potential to activate in circulation or not to be fully quenched by circulating protease inhibitors. I am concerned that this could result in damage outside the pancreas-careful review of lung, splenic, and liver pathology is needed.
6. If tissue is examined prior to overt pathologic injury, is there evidence of an ER-stress response?
7. How does the activation of trypsinogen in an acinar cell lead to pancreatic damage? Are hallmarks of acute acinar cell changes observed in acute pancreatitis present?
8. Does the pancreatic duct epithelium develop PanIN-like lesions late in disease (as described in human tissues)?
9. The authors should cite/discuss the Amer J Surg Pathol March 2014 paper by AD Singhi et al on the histopathology of PRSS1 pancreatitis and the relationship of their findings to this work.

Reviewer #2 (Remarks to the Author):

The manuscript entitled "A preclinical model of chronic pancreatitis driven by trypsinogen activation" by Geisz and Sahin-Toth describe the development and characterization of a new genetic engineered mouse model of spontaneous acute and chronic pancreatitis. The model was developed based on compelling arguments from over 20 years of human genetic association studies as well as laboratory work on the trypsinogen pathway. The model is well designed and relevant to human disease mechanism.

Many other groups have attempted to design genetic models of human hereditary pancreatitis in mice and rats, but none have been successful for a variety of technical reasons. It appears that Geisz and Sahin-Toth have been successful in a model described in this report.

The manuscript does a nice job of describing the rationale, approach, development and testing of the GEMM. The supplemental information is useful to people in the field, and the main document lays out the important features of the model in a clear and logical presentation. The results and interpretation appear to be expected and valid.

A few criticisms of an otherwise compelling story:

- 1) The title uses the term "preclinical" model. However, this designation is premature, as the manuscript only describes the characteristic of a new GEMM and does not include any intervention that could be described as "preclinical"
- 2) The authors should review the use of the protein and gene symbols to be sure they are properly italicized and capital or lower case according to convention and style of the journal.

Reviewer #3 (Remarks to the Author):

It was a pleasure to review this piece of work by Andrea Geisz and Miklós Sahin-Tóth . The senior author is a towering figure in the field of pancreatitis and he has correctly identified the need for a preclinical model of chronic pancreatitis.

A GEM driven by trypsinogen auto-activation is an obvious approach but has not been particularly well achieved until perhaps this model. Certainly what is needed in the Introduction and Discussion is mention of other types of murine models for chronic pancreatic including those involving chemical stimulation and/or genetic modification. All of the current models have some disadvantages and these need to be described.

The Mechanistic Definition provides a benchmark against which this model needs to be evaluated (DC Whitcomb et al 2016): " 'Chronic pancreatitis is a pathologic fibro-inflammatory syndrome of the pancreas in individuals with genetic, environmental and/or other risk factors who develop persistent pathologic responses to parenchymal injury or stress.' In addition, "Common features of established and advanced CP include pancreatic atrophy, fibrosis, pain syndromes, duct distortion and strictures, calci fications, pancreatic exocrine dysfunction, pancreatic endocrine dysfunction and dysplasia." "

Whilst this model may be a significant advance it needs to be better described; and where it might fall down on expectations then this also needs to be mentioned.

What we have is a GEM developing inflammatory changes very early, then associated with some fibrosis and later atrophy with adipose deposition but preservation of acini.

This does happen in the human setting but only in a small minority of cases.

People with chronic pancreatic usually have an insidious onset that may or may not be punctuated by attacks of acute pancreatitis. Once established what we see is a parenchymal architecture disruption of the normal acinar lobules and a combination of acinar loss , collagen deposition, inflammatory infiltrate and duct distortion. This process tends to continue ultimately resulting in complete architectural destruction with a predominance of fibrosis, acinar loss and also loss of islets. The alternative pathway is acinar atrophy.

In the current model the authors could provide additional information as follows:

1. Confirm that the T7D23A knock in was stable beyond four weeks when trypsinogen auto activation was at its peak.
2. Quantify the amount the collagen.
3. Quantify activated pancreatic stellate cells.
4. Quantify the inflammatory cell infiltrate.
5. Quantify the number of islet cells.
6. If the colony still exists it would be interesting to see if isolated acini are more sensitive to pathophysiological signals.

Do the authors plan to create an inducible model? This might be more relevant, allowing the abnormal gene to be activated when the mice are older and also determine the interaction with patho-supra-physiological stimuli, such as alcohol and certain bile acids.

John Neoptolemos, Heidleberg

NCOMMS-18-12687-T

“A preclinical model of chronic pancreatitis driven by trypsinogen autoactivation”

Point-by-point response to the referees’ comments

Reviewer #1.

The submission by Geisz and Sahin-Toth characterizes a new murine model of chronic pancreatitis that is designed to parallel mechanistically and pathologically the human disease known as Hereditary Pancreatitis. This genetic etiology of this disease is mutation of the human cationic gene; to recapitulate this disease in mouse, the authors have make a knock-in a murine trypsinogen isoform that they predict will reflect the defect that causes human disease. Previous attempts to generate such a mouse model have had limited success for a variety of reasons. This model is proposed to cause disease by enhancing the autoactivation of the trypsinogen. The authors show that the mutation induces prominent pancreatic disease; further characterization is needed to determine to what extent it parallels human disease. The input of a pathologist familiar with the pancreatitis, especially the histologic features of Hereditary Pancreatitis as well as murine models of pancreatic disease would be useful.

1. The authors state that the earliest changes in pathology are those of acute pancreatitis. However, the pathology shown appears not to show acute inflammation, a central feature of acute pancreatitis. Indeed, the inflammation appears to be primarily mononuclear. As part of the initial characterization of the model, the authors should label the tissue for inflammatory cell subtypes (neutrophils, lymphocytes, and macrophages) and quantify the infiltrates at various disease stages. If lymphocytes are abundant, particularly at the very early stages of disease, they should be subtyped (at least as T or B categories). Since the authors propose this model might be used for clinical trials, is it critical to establish the similarity (and differences from human disease).

We thank the Referee for this important suggestion. We performed immunohistochemistry (IHC) on pancreas sections from *T7D23A* mice with acute pancreatitis for myeloperoxidase (MPO) to label neutrophils and F4/80 to label macrophages. We obtained strong labeling for both markers. We compared this pattern of labeling with that of cerulein-induced acute pancreatitis (performed on control mice); a benchmark model of murine experimental pancreatitis. Again, strong labeling was observed for both markers, indicating that inflammatory cell infiltrates are of mixed nature in this model as well. In contrast to the MPO and F4/80 markers, we found essentially no labeling with the T and B lymphocyte markers CD3 and CD45R/B220; excluding significant lymphocyte infiltration. We added these new IHC data to the revised manuscript as Figure 3. The cerulein-induced pancreatitis IHC data we submitted for the referee’s review only (Figure 1 for review only).

In addition, we used an ELISA to measure MPO in pancreas homogenates of *T7D23A* mice with acute pancreatitis (aged 3.5 weeks) and compared these values to those of untreated (control) and cerulein-treated C57BL6/N mice. MPO values were markedly higher in *T7D23A* animals (see Figure 1 for review only).

2. In the context of #1, the dramatic mononuclear response, especially at the early stage of disease, seems very unusual. I wonder if the authors have generated a form of lymphocytic pancreatitis and if the mutated trypsinogen could be a neo-antigen. In this context, the authors should compare serum from WT, early, and late time period mutants (several animals at each age), for antibodies to general pancreatic determinants (homogenates) as well as to the mutant protein.

This is an exciting idea; however, this does not seem to be the case. As described above under point #1, there are no lymphocytic infiltrates present. Furthermore, now we have a new mouse model, which carries not only the D23A mutation in T7 trypsinogen but also a mutation in the activation site (K24G). This trypsinogen cannot undergo autoactivation. These mice do not exhibit any signs of pancreatic pathology (so far followed up to 5.5 months). This control strain rules out alternative disease mechanisms associated with the D23A mutation, such as trypsinogen misfolding or neoantigen formation and confirms the pathogenic role of increased trypsinogen autoactivation. In light of these new data, we believe that the suggested immunological experiments are unnecessary and we hope the Referee will concur with our assessment.

3. In that context, an elevation in serum levels of amylase does not necessarily mean that pancreatitis is present. For example, changes in duct permeability and/or basolateral exocytosis could cause similar findings. If there are no neutrophils to accompany the elevations in amylase, this reviewer would caution about the pathologic term acute pancreatitis, even if the amylase is elevated.

Again, we are grateful to the Referee for stressing this point, which is related to his comment under point #1. Since we demonstrated ample neutrophil infiltration, we are confident that we can call the pathology observed acute pancreatitis. We note, furthermore, that our *T7D23A* mice also satisfy the 2/3 clinical criteria established for the diagnosis of acute pancreatitis in humans. There is significant serum amylase elevation and pancreatic edema (measured directly here and by imaging in humans). Finally, we agree with the Referee regarding the potential mechanisms of serum amylase increase during acute pancreatitis. It is still debated whether acinar cell damage versus altered basolateral secretion underlies the characteristic serum amylase increase.

4. One of the hallmarks of Hereditary pancreatitis is intraductal calcifications-some images suggest these may be present-it would be of value for the authors to review the pancreatic pathology with a pathologist, with one goal of being to state whether there are intraductal calcifications. Moreover, some centers have high resolution ultrasonography and/or computed tomography for mice. It would be very useful to show pancreatic calcifications by imaging in addition to pathology.

Excellent point. We performed von Kossa staining on sections from 2, 6 and 12 month-old *T7D23A* animals and found no indication of intraductal or interstitial calcifications (see Figure 2 for review only). We added this information to the revised manuscript.

We note that bicarbonate peak concentrations in pancreatic ducts are much lower in mice (~50-70 mM) than in humans (>140 mM). That is the reason why ductal studies use guinea pigs as a model

animal, which is capable of secreting up to 140 mM bicarbonate. Due to this difference in ductal physiology between mice and humans, we do not expect to see intraductal stones in mice.

5. The mutant protein has a potential to activate in circulation or not to be fully quenched by circulating protease inhibitors. I am concerned that this could result in damage outside the pancreas-careful review of lung, splenic, and liver pathology is needed.

We reviewed hematoxylin-eosin stained sections of the lung, liver, spleen and kidney from *T7D23A* mice (aged 6 weeks). They all looked normal when compared to sections from control C57BL/6N mice. We added a statement describing these observations to the revised manuscript and submitted the pictures as Figure 3 for review only.

6. If tissue is examined prior to overt pathologic injury, is there evidence of an ER-stress response?

We thank the Referee for this interesting suggestion. We measured mRNA expression for the ER master chaperone Hspa5 (BiP) and pro-apoptotic ER-stress related transcription factor Ddit3 (CHOP) in the pancreas from *T7D23A* and control C57BL/6N mice aged 3 weeks, 4 weeks and 5 weeks. As described in the manuscript, most of the 3 week-old mice have no pathology, whereas 4 and 5 week old mice have acute or early chronic pancreatitis. With the exception of two mice (aged 4 weeks), we found no significant changes in BiP expression. In contrast, significantly elevated CHOP expression was observed in mice aged 4 or 5 weeks, whereas at 3 weeks the majority of mice showed no increase. We decided not to include these data in the revised manuscript as we feel that interpretation is difficult due to the altered tissue composition of the diseased pancreas at 4 and 5 weeks. The most reliable conclusion from these experiments is that ER stress is absent or minimal in the pancreas of *T7D23A* mice before pancreatitis onset. We submitted these data as Figure 4 for review only.

7. How does the activation of trypsinogen in an acinar cell lead to pancreatic damage? Are hallmarks of acute acinar cell changes observed in acute pancreatitis present?

This is a question our entire field has been debating for decades. According to the latest model from the Saluja group, trypsin-induced leakage of cathepsin B is responsible for acinar cell injury (Gastroenterology 2016; 151:747-758). In contrast, the Lerch group claims a direct damaging effect of trypsin on acinar cells (J Biol Chem 2016; 291:14717-14731). We hope our model will bring novel insight in this regard. With respect to acinar cell changes during acute pancreatitis, we observed consistent and significant necrosis in the center of the lobules (see Figure 2B). This is different from the necrosis seen in the cerulein-induced model, which tends to appear at the lobule periphery. We also see scattered acinar cell vacuolization. Future studies using isolated acinar cells from *T7D23A* mice may be even more informative with respect to the sequence of events from trypsin activation to acinar cell damage.

8. *Does the pancreatic duct epithelium develop PanIN-like lesions late in disease (as described in human tissues)?*

In mice aged 6-12 months, we did observe sporadic development of early PanIN-like lesions. This is a relatively rare occurrence, which we still actively study. Unfortunately, the extended timeline and the low incidence of these changes hamper our efforts in this regard. We submitted pictures of the lesions for the referee's review only (Figure 5 for review only).

9. *The authors should cite/discuss the Amer J Surg Pathol March 2014 paper by AD Singhi et al on the histopathology of PRSS1 pancreatitis and the relationship of their findings to this work.*

This is indeed a seminal paper highly relevant to our study. We were well aware of this article and we often discussed it in the lab. We feel embarrassed for omitting this. Now we included the citation in our revised manuscript and we briefly discuss the similarities to our model.

Reviewer #2.

The manuscript entitled "A preclinical model of chronic pancreatitis driven by trypsinogen activation" by Geisz and Sahin-Toth describe the development and characterization of a new genetic engineered mouse model of spontaneous acute and chronic pancreatitis. The model was developed based on compelling arguments from over 20 years of human genetic association studies as well as laboratory work on the trypsinogen pathway. The model is well designed and relevant to human disease mechanism.

Many other groups have attempted to design genetic models of human hereditary pancreatitis in mice and rats, but none have been successful for a variety of technical reasons. It appears that Geisz and Sahin-Toth have been successful in a model described in this report.

The manuscript does a nice job of describing the rationale, approach, development and testing of the GEMM. The supplemental information is useful to people in the field, and the main document lays out the important features of the model in a clear and logical presentation. The results and interpretation appear to be expected and valid.

A few criticisms of an otherwise compelling story:

1. *The title uses the term "preclinical" model. However, this designation is premature, as the manuscript only describes the characteristic of a new GEMM and does not include any intervention that could be described as "preclinical".*

Technically the Referee is correct. However, we view the primary utility of this exciting new model as a preclinical tool, which was reflected in the title. We wish to keep the title in its original form and we respectfully request the Referee to concur with our position.

2. *The authors should review the use of the protein and gene symbols to be sure they are properly italicized and capital or lower case according to convention and style of the journal.*

This has been reviewed and corrected where necessary.

Reviewer #3.

It was a pleasure to review this piece of work by Andrea Geisz and Miklós Sahin-Tóth . The senior author is a towering figure in the field of pancreatitis and he has correctly identified the need for a preclinical model of chronic pancreatitis.

A GEM driven by trypsinogen auto-activation is an obvious approach but has not been particularly well achieved until perhaps this model. Certainly what is needed in the Introduction and Discussion is mention of other types of murine models for chronic pancreatic including those involving chemical stimulation and/or genetic modification. All of the current models have some disadvantages and these need to be described.

We thank the Referee for suggesting this additional point of discussion. We reviewed all previously published GEM models of trypsin-dependent pancreatitis in the Introduction of our revised paper.

The Mechanistic Definition provides a benchmark against which this model needs to be evaluated (DC Whitcomb et al 2016):” ‘Chronic pancreatitis is a pathologic fibro-inflammatory syndrome of the pancreas in individuals with genetic, environmental and/or other risk factors who develop persistent pathologic responses to parenchymal injury or stress.’ In addition, “Common features of established and advanced CP include pancreatic atrophy, fibrosis, pain syndromes, duct distortion and strictures, calci fications, pancreatic exocrine dysfunction, pancreatic endocrine dysfunction and dysplasia.””

We are grateful for this suggestion. We added discussion on “Comparing features of T7D23A mice to those of human chronic pancreatitis”. In this paragraph, we followed the Referee’s suggestion and used the 1996 Whitcomb et al. Pancreatology paper as a benchmark.

Whilst this model may be a significant advance it needs to be better described; and where it might fall down on expectations then this also needs to be mentioned.

What we have is a GEM developing inflammatory changes very early, then associated with some fibrosis and later atrophy with adipose deposition but preservation of acini.

This does happen in the human setting but only in a small minority of cases.

People with chronic pancreatic usually have an insidious onset that may or may not be punctuated by attacks of acute pancreatitis. Once established what we see is a parenchymal architecture disruption of the normal acinar lobules and a combination of acinar loss , collagen deposition, inflammatory infiltrate and duct distortion. This process tends to continue ultimately resulting in complete architectural destruction with a predominance of fibrosis, acinar loss and also loss of islets. The alternative pathway is acinar atrophy.

In the current model the authors could provide additional information as follows:

- 1. Confirm that the T7D23A knock in was stable beyond four weeks when trypsinogen autoactivation was at its peak.*

We confirmed this by sequencing cDNA from the pancreas of T7D23A mice at various ages. We always found two comparable signals at the site of the mutation (as shown in Figure 1E), indicating comparable expression of the wild-type and mutant T7 trypsinogen alleles.

2. Quantify the amount the collagen.

We measured the hydroxyproline content of the pancreas. We included these new data in the revised manuscript as Figure 4B.

3. Quantify activated pancreatic stellate cells.

We performed immunohistochemistry for alpha-smooth muscle actin to label activated stellate cells in the pancreas of T7D23A mice, aged 4 weeks. As expected, we obtained widespread labeling while pancreas tissue from control C57BL/6N mice was not labeled. We added these new data to the revised manuscript Figure 4C.

4. Quantify the inflammatory cell infiltrate.

Please see our response to Referee #1, point #1.

5. Quantify the number of islet cells.

We performed qualitative evaluation of islet-cell morphology, number and function. In the end-stage pancreas with extensive adipose replacement, one still can see abundant, normal-looking islets, which appear slightly enlarged (see Figure 2D). With respect to islet function, we measured blood glucose levels in 12 months old animals and found normal concentrations (see data in revised manuscript). We agree that our model offers a unique opportunity to study islet cell function in the context of chronic pancreatitis, but these studies are beyond the scope of the present paper.

6. If the colony still exists it would be interesting to see if isolated acini are more sensitive to pathophysiological signals.

This is an excellent suggestion. We are trying to establish methodology for the isolation of pancreatic acini from very young mice, which are still free from disease pathology. We hope we can address this question in the near future. As an alternative approach, we are developing a mouse strain, which expresses the same trypsinogen mutant at lower levels and exhibits later onset of pancreatic disease. This would allow us to test the question the Referee raised either by directly stimulating mice with cerulein or by isolating acini from adult animals.

Do the authors plan to create an inducible model? This might be more relevant, allowing the abnormal gene to be activated when the mice are older and also determine the interaction with patho-supra-physiological stimuli, such as alcohol and certain bile acids.

Yes, we will pursue this option as soon as funding permits. In our original design, we wished to model the human situation where carriers are born with heterozygous mutation in their trypsinogen gene. We agree that for certain mechanistic studies an inducible model would be preferred.

REVIEWERS' COMMENTS:

Reviewer #1 (Remarks to the Author):

The authors present convincing data showing that there is a prominent non-lymphoid infiltrate in their model. They also provide new convincing evidence that the model develops some of the classic features of chronic pancreatitis (fibrosis and dropout). Modification in the text and addition of key citations also enhanced the text. The work should be of broad interest.

Reviewer #3 (Remarks to the Author):

The authors have done well in addressing many but not all of the issues.

I can understand why they have not undertaken an inducible model.

I would have preferred to see isolated acinar responses in vitro to physiological and pathophysiological stimuli.

Not doing this weakens the authenticity of the model.

In the discussion they make contradictory statements as to whether or not this is a true model of human chronic pancreatitis.

This model tends to parenchymal atrophy with adipose tissue replacement and islet cell preservation, whereas the human form tends to severe fibrosis, preservation of nerve fibres and loss of islets as well as acini.

This difference is not properly explained - the potential weakness of the model needs to be mentioned in the discussion.

Why is there relatively little fibrosis despite extensive stellate cell activation?

The model is nevertheless a step in the right direction

NCOMMS-18-12687-A

“A preclinical model of chronic pancreatitis driven by trypsinogen autoactivation”

Point-by-point response to the referees' comments

Reviewer #3 (Remarks to the Author):

The authors have done well in addressing many but not all of the issues. I can understand why they have not undertaken an inducible model. I would have preferred to see isolated acinar responses in vitro to physiological and pathophysiological stimuli. Not doing this weakens the authenticity of the model.

We agree with the Reviewer. As we discussed in our previous responses, isolation of acini is problematic due to the rapid onset of destructive disease in this model. We are working on new models with later onset of pancreatitis, which would allow us to isolate acini before the disease develops.

In the discussion they make contradictory statements as to whether or not this is a true model of human chronic pancreatitis. This model tends to parenchymal atrophy with adipose tissue replacement and islet cell preservation, whereas the human form tends to severe fibrosis, preservation of nerve fibres and loss of islets as well as acini. This difference is not properly explained - the potential weakness of the model needs to be mentioned in the discussion.

To clarify this point better, we added the following statement to the discussion: “Although some fibrosis is seen early on, ultimately adipose replacement rather than severe fibrosis dominates the histopathology”.

Why is there relatively little fibrosis despite extensive stellate cell activation?

This is an interesting point. We cannot be certain but we speculate that the process of adipose replacement suppresses stellate cell activity and limits fibrosis.

The model is nevertheless a step in the right direction.

We appreciate this and we could not agree more.